

# A systematic review of conference papers presented at two large Japanese psychology conferences in 2013 and 2018: did Japanese social psychologists selectively report $p < 0.05$ results without peer review?

Kai Hiraishi[1], Asako Miura[2], Masataka Higuchi[3], Yoshitsugu Fujishima[4], Daiki Nakamura[5] and Masaki Suyama[6]

[1] Faculty of Letters, Keio University, Minatoku, Tokyo, Japan
[2] Graduate School of Human Sciences, Osaka University, Suita, Osaka, Japan
[3] Faculty of Human Sciences, Sophia University, Chiyodaku, Tokyo, Japan
[4] Faculty of Humanities and Social Sciences, Showa Women's University, Setagayaku, Tokyo, Japan
[5] Faculty of Education, University of Miyazaki, Miyazaki, Miyazaki, Japan
[6] Faculty of Psychology, Yasuda Women's University, Asaminamiku, Hiroshima, Japan

Corresponding author
Kai Hiraishi, kaihiraishi@keio.jp

## ABSTRACT

We conducted a systematic review of conference papers in social psychology at two large psychology conferences in Japan: the Japanese Psychological Association and the Japanese Society for Social Psychology. The conference papers were effectively not subjected to peer review; hence, they were suitable for testing if psychologists selectively reported statistically significant findings without pressure from journal editors and reviewers. We investigated the distributions of $z$-values converted from the $p$-values reported in the articles presented at the 2013 and 2018 conferences. The $z$-curve analyses suggest the existence of selective reporting by the authors in 2013. The expected discovery rate (EDR) was much lower than the observed discovery rate (ODR; 7% *vs*. 76%, respectively), and the 95% confidence interval (CI) did not include the ODR. However, this does not mean that the set of studies completely lacked evidential value. The expected replication rate (ERR) was 31%; this is significantly higher than 5%, which was expected under the null hypothesis of no effect. Changes were observed between 2013 and 2018. The ERR increased (31% to 44%), and the EDR almost doubled (7% to 13%). However, the estimation of the maximum false discovery rate (FDR; 68% in 2013 and 35% in 2018) suggested that a substantial proportion of the reported findings were false positives. Overall, while social psychologists in Japan engaged in selective reporting, this does not mean that the entire field was covered with false positives. In addition, slight signs of improvement were observed in how they reported their findings. Still, the evidential value of the target studies was weak, even in 2018, allowing for no optimism.

## INTRODUCTION

### Background

It has been almost 10 years since psychologists discovered the replicability crisis in their discipline. Over the years, the surprisingly (or not surprisingly) low replicability rates of psychological studies published in prestigious journals have been documented (*Open Science Collaboration, 2012*, *2015*). Questionable research practices (QRPs) and *p*-hacking behind the crisis have also been identified (*Simmons, Nelson & Simonsohn, 2011*). It is argued that scientifically inappropriate conduct by psychologists is the product of publication biases in favor of novel and statistically significant (*i.e.*, $p < 0.05$) results. The logic is as follows: Researchers need more publications in prestigious journals to be promoted. Psychological journals prioritize the publication of novel, surprising, and statistically significant studies. As such, authors have no choice but to resort to QRPs to squeeze publishable $p < 0.05$ results out of their data. Due to poor synergy, studies with low replicability have accumulated in the field.

However, are journal editors and reviewers solely responsible for this crisis? The answer can hardly be yes because they are also members of the academic community. It is more probable that the psychological community has (or used to have) a culture that values novel, surprising, and statistically significant findings and that editors and reviewers follow cultural norms when evaluating submitted manuscripts. There is no denying that such actions by editors and reviewers have worsened the scientific integrity of the discipline. Nevertheless, we suspect that it is not only journal editors and reviewers, but also psychologists in general, who value novelty and surprisingness too much. If our conjecture is correct, psychologists might tend to selectively report novel, surprising, and statistically significant results even in the absence of pressure from journal editors and reviewers. We tested part of this hypothesis in the field of social psychology.

We set our target to social psychology primarily because for most of us, social psychology is our area of expertise. In addition, we believe there is a specific social value in studying the evidential value of social psychology. Publication bias and excessive publication of statistically significant results have been documented in many subfields of social science, including experimental economics (*Brodeur, Cook & Heyes, 2020*; *Maier et al., 2022b*), experimental social science (*Franco, Malhotra & Simonovits, 2014*), and social psychology (*e.g.*, *Carter & McCullough, 2014*; *Chen et al., 2022*; *Maier et al., 2022a*; *Schimmack, 2020*). Further, social psychology is infamous for its low replicability compared to other fields in psychology (*Open Science Collaboration, 2015*). This situation has sparked controversy over the applicability of social science findings to real-world problems such as the COVID-19 pandemic (*IJzerman et al., 2020*; *Van Bavel et al., 2020*). As social psychology constitutes a significant part of social science, elucidating the mechanism underlying the problems should also have significant social value.

## Peculiarities of the conference paper format of Japanese psychological societies

We propose that the annual conferences of two Japanese academic societies, the Japanese Psychological Association (JPA) and the Japanese Society for Social Psychology (JSSP), provide interesting and unique resources to test the hypothesis; social psychologists prioritize statistical significance regardless of reviewer preferences. With more than 1,000 members (about 8,000 for the JPA and 1,700 for the JSSP), the two societies are among the largest academic communities of psychology researchers in Japan. The two societies' main presentation formats at the annual conferences were poster presentations. Each year, hundreds of posters are presented at conferences; the posters at these conferences are one of the main research outlets for Japanese social psychologists. For instance, there were 254 poster presentations at the JSSP conference in 2017, whereas only 18 articles were published in the society's official journal.

There are two peculiar characteristics of the poster format. First, the authors of a poster must submit a two-column, one-page conference paper in A4 format. The authors can (and must) report the details of their study's hypotheses, methods, and results. While the posters are presented only at the conference venue, conference papers are archived and made publicly available online by the societies. These conference papers are often included in researcher's curriculum vitae and are used to evaluate their annual achievements.

Second, the conference papers do not undergo a typical review process. The conference committee only checks if the papers meet the minimum requirements as scientific papers and does not reject them based on the scientific value of the research. Specifically, they examine whether the paper is well formatted (*e.g.*, it does not exceed a one-page limit) and they verify that the study does not have serious ethical violations or conflicts of interest. There is no further peer review evaluating the paper's scientific value. As a result, almost all submissions are accepted for presentation at the venue, and the conference papers are archived and published. Therefore, authors do not have to care about the peer reviewers' evaluation of the novelty, surprisingness, or statistical significance of their study.

The conference papers of the two societies combined constitute an archive of what Japanese social psychologists have been doing when they can publicize their studies without peer review. We aimed to systematically review the accumulated records of psychologists' behavior.

## Purpose of the systematic review

This systematic review proceeded in two stages. In the first stage, we explored the details of the conference papers (see Supplemental 1 for details). We checked and coded whether each paper reported details of the hypotheses, predictions, methods, and results, such as sample sizes, $p$-values, means, standard deviations (SDs), test statistics, effect sizes, and confidence intervals (CIs). Given this information, we proceeded to the second stage and examined the evidential value of the studies presented at the JPA and the JSSP. Specifically, we collected the exact statistical values—such as $p$-values, $t$-values, $F$-values, and degrees of freedom (DFs)—from conference papers (Supplemental 2). This procedure enabled us to conduct $p$-curve analyses (*Simonsohn, Nelson & Simmons, 2014a*, *2014b*; *Simonsohn,*
*Simmons & Nelson, 2015*; *van Aert, Wicherts & van Assen, 2016*) and *z*-curve analyses (*Bartoš & Schimmack, 2022*; *Brunner & Schimmack, 2020*).

The *p*- or *z*-curve analyses are meta-analytic techniques that rely solely on the reported *p*-values.[1] With *p*- or *z*-curve analyses, we can estimate the evidential value of a set of target studies. They examine how well the distribution of the reported *p*-values (or *z*-values converted from the *p*-values) matches the expected distribution under specific assumptions. For instance, the distribution of *z*-values converted from the *p*-values should follow a normal distribution with mean = 0 and *SD* = 1 if the effect in question is null. Either by comparing the expected *p*-curve and the observed distribution of *p*-values (*p*-curve analysis) or by finding a mixture of normal distributions (to be precise, a mixture of truncated folded normal distributions with differing means) that best fits the distribution of the observed *z*-values (*z*-curve analysis), these techniques estimate the evidential value of a set of studies. In addition, the latest version of *z*-curve analysis (*z*-curve 2.0) can estimate the percentages of statistically significant studies among all studies that have ever been conducted (*Bartoš & Schimmack, 2022*). This enables us to directly examine the existence and magnitude of publication bias. Using *z*-curve analysis, several studies have revealed significant publication bias in social psychology research (*Sotola & Credé, 2022*; *McAuliffe et al., 2021*; *Maier et al., 2022a*; *Schimmack, 2020*).

It should be noted that *p*- or *z*-curve analyses do not directly reveal the existence of QRPs or *p*-hackings. Nevertheless, we believe we can indirectly examine the appropriateness of the researcher's behavior by estimating the evidential value of conference papers. If most authors have resorted to QRPs, this should inevitably produce a set of studies with low evidential value (*e.g.*, low expected probabilities of successful replication). In the current article, we mainly report the results of *z*-curve analyses, whereas we pre-registered to conduct *p*-curve analysis as well. This is because it became apparent during the review process that *p*-curve analysis is redundant for the current study (see Supplemental 4 for the *p*-curve results).

In the current study, we did not plan to compute the aggregated effect sizes of the target articles, although this was possible with the *p*/*z*-curve analyses. Due to the rather wide variations in topics and fields of the target papers, we posited that estimating the "aggregated" effect size would be almost meaningless.

## Target studies

We reviewed social psychology conference papers presented at either the JPA or the JSSP in 2013 and 2018. A large number of papers were presented at these two conferences each year. Hence, we limited the target year of presentation to a fairly distant year (2013), when Japanese social psychologists did not share sufficient knowledge about the replicability crisis, and to a more recent year (2018). During the 5 years between 2013 and 2018, several attempts were made to advertise the crisis to Japanese psychologists; these included the publication of special issues on the replicability crisis in the journal *Japanese Psychological Review* (*Miura, Okada & Shimizu, 2018*; *Miura et al., 2019*; *Tomonaga, Miura & Haryu, 2016*) and holding symposia at several academic conferences, including the JPA and JSSP.

[1] Strictly speaking, these two techniques differ from traditional meta-analysis. Whereas traditional meta-analysis aims to estimate an average effect size across the studies included, the main purpose and strength of *p*-curve and *z*-curve analyses do not lie in such an estimation.

**Table 1 Number of studies categorized as eligible at each stage.**

| | | 2013 | | 2018 | |
|---|---|---|---|---|---|
| | Decision nodes | Yes | % | Yes | % |
| | All papers | 689 | | 499 | |
| Stage 1 | Experiment?[a] | 210 | 30% | 162 | 32% |
| | Hypothesis testing?[b] | 159 | 23% | 128 | 26% |
| Stage 2 | p-stats?[c] | 91 | 57% | 66 | 52% |
| | significant?[d] | 71 | 45% | 42 | 33% |
| | p-value/stats?[c] | 97 | 61% | 77 | 61% |
| | significant?[d] | 74 | 47% | 49 | 39% |

**Notes:**
[a] Number of papers reporting one or more experiments
[b] Number of papers with one or more directed hypotheses
[c] Number of papers reporting $p$-value(s) and/or $p$-stats for the directed hypothesis
[d] Number of studies with significant result(s) ($p < 0.05$). Pecentages indicate the ratio of studies that corresponded to the criterion among the studies that passed the earlier decision node. For instance, the number of significant studies with $p$-stats in 2013 (71) was 45% of the studies that were testing directed hypotheses with an experiment (159).

As such, we conjectured that some changes may have occurred during the years in terms of how social psychologists reported their studies.

In the early stages of review protocol development, we found that there was considerable heterogeneity in the quantity and quality of information written in each conference paper. For instance, the statistics reported in papers varied greatly depending on whether the study was a questionnaire survey or an experiment. Hence, it was unrealistic to construct a one-size-fits-all review protocol. As such, we chose to focus only on experimental studies, most of which employed analysis of variance (ANOVA) and were relatively easy to cover under one protocol. Among the conference papers reporting social psychological experimental studies, we included those that described details of statistical values sufficient to conduct $p$- and $z$-curve analyses.

## METHODS

### Transparency and openness

We pre-registered the review protocols and analysis plans in an open-science framework. Data and R scripts for the study are available at: https://osf.io/kdztx.

### Eligibility criteria and coding procedures

We first identified all conference papers accompanying poster presentations at the JSSP annual conferences in 2013 and 2018 from the society's archive (https://iap-jp.org/jssp/conf_archive). For the JPA, papers accompanying the posters in the "Society and Culture" section for the 2013 and 2018 conferences were identified and downloaded from the archive (www.jstage.jst.go.jp/browse/pacjpa/-char/en). There were 689 and 499 papers in 2013 and 2018, respectively (see Table 1).[2]

In the first coding stage, we selected papers that met two eligibility criteria: (1) whether the study was experimental, and (2) the study presented directed predictions in the introduction (Fig. 1). By "experimental," we mean that the researchers controlled for at

[2] Although we had pre-registered only poster presentations as eligible, due to technical difficulties, we included both oral and poster presentations in the eligible papers for the JSSP conference. There were 153 and 233 oral presentations in 2013 and in 2018, respectively. Authors of oral presentations submitted the same A4 size conference paper as authors of poster presentations.

least one factor. As such, a study with only naturally occurring factors (*e.g.*, culture, gender) was not coded as "experimental" no matter how complex the techniques or technologies that the researchers employed (*e.g.*, IAT, fMRI). Meanwhile, we coded vignette studies as "experimental" as long as the researchers manipulated one or more factors.

For each experimental study, we coded whether it had directed prediction(s) for the effects that corresponded to the research questions. For instance, in a 2 (X1a *vs.* X1b) × 2 (X2a *vs.* X2b) design, there are three effects (two main effects and one interaction effect), each of which can be a target of the research question. If the paper explicitly stated a prediction for the main effect of X1 (*e.g.*, X1a > X1b), it was coded as having a directed prediction. If the paper only proposed that there could be a difference between X2a and X2b, but no clear statement about the direction of the effect, we coded it as non-directed. For the interaction term, if the study predicted either attenuation (*e.g.*, X2 attenuates the simple effect of X1 on Y) or reversal of the effects (X1a > X1b for X2a and X1a < X1b for X2b), we coded it as directed. If the author stated nothing regarding an effect, we coded it as having no prediction.

The second coding stage targeted only those papers with at least one experimental study and at least one directed prediction. We collected detailed statistical information sufficient to compute the *p*-value (*e.g.*, the *t*-value and the degrees of freedom for a *t*-test, hereafter *p*-stats), *p*-values, or both for a directed prediction. When multiple studies with multiple effects were reported in one conference paper, we collected the *p*-stats and/or *p*-value that first appeared in the paper's main text for the primary analyses. We also collected the second and subsequent *p*-stats and *p*-values for sensitivity analyses. Papers that reported only the alpha level (*e.g.*, $p < 0.05$) were not eligible for the current systematic review.

We developed the data collection protocol based on several pilot collections. The first three pilot collections and coding were conducted by the initial members of the project (KH, AM, MH, and YF), who designed the alpha version of the Stage 1 protocol. Two coders (MS and DN), the co-authors of the current study, performed two additional pilot selections and coding. The beta version was proposed accordingly through discussion among the authors. The official version of the Stage 1 protocol was finalized by carrying out selection and coding on 100 papers randomly selected from the target papers in 2018 (499 papers from the JPA and JSSP in total). Any discrepancies between the two coders at this stage were discussed, resolved, and reflected in the pre-registered version of the protocol. Two coders split the remaining 399 papers in half and conducted the selection and coding. At this stage, the coders started collecting *p*-stats and *p*-values alongside other information. Again, several discrepancies between the two coders were observed and resolved. This resulted in Stage 1, Revision 1 of the protocol (Supplemental 1). Subsequently, the coders began selecting and coding papers in 2013 by splitting the 690 papers (the JPA and JSSP in total) in half.

### Analysis plan

We planned to conduct *p*-curve (*Simonsohn, Nelson & Simmons, 2014a*, *2014b*; *Simonsohn, Simmons & Nelson, 2015*) and *z*-curve analyses (*Bartoš & Schimmack, 2022*;

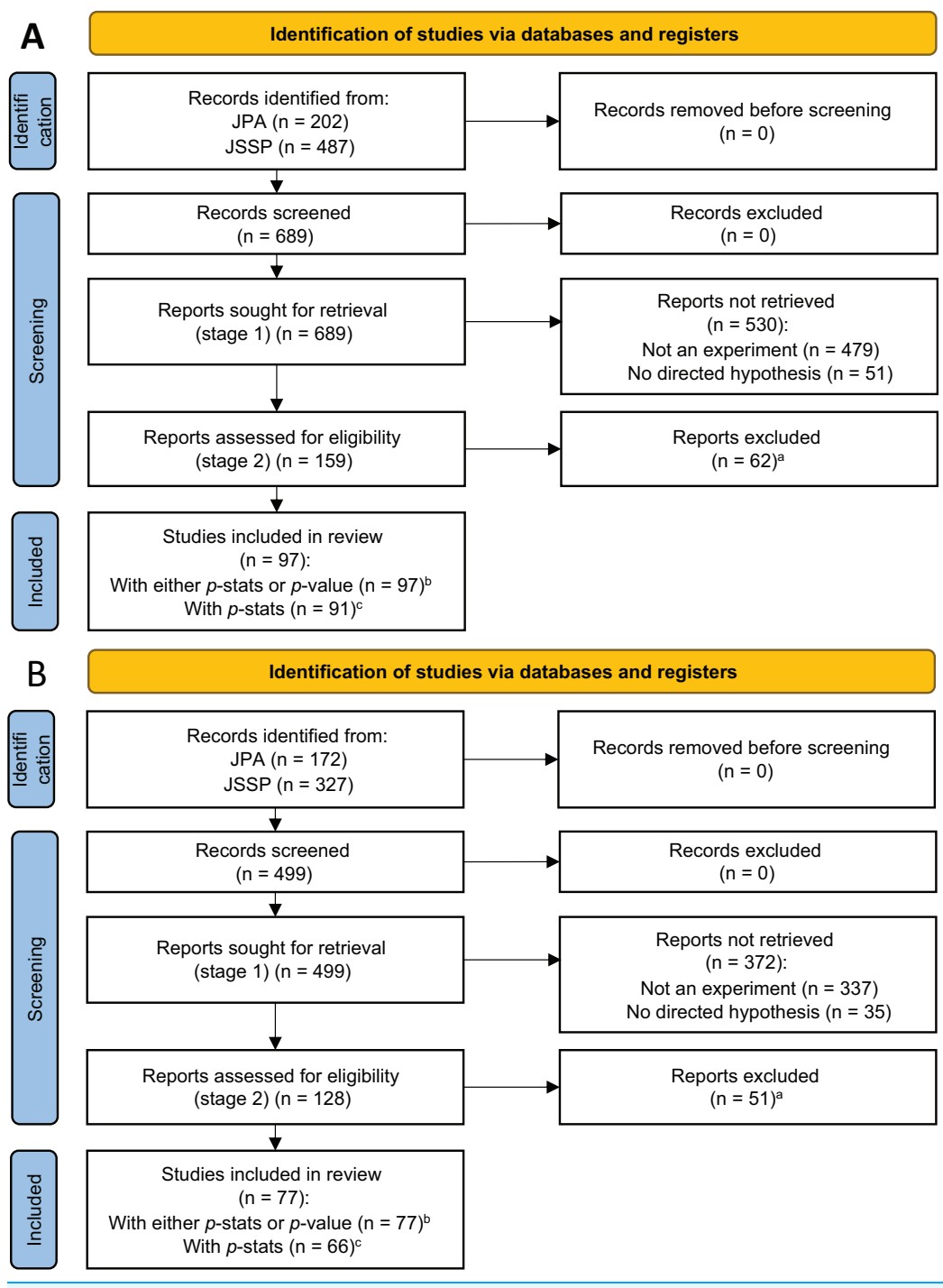

**Figure 1  PRISMA flow diagram for studies in 2013 (A) and 2018 (B).** (A) Reports without valid *p*-stats or *p*-value were excluded. In 2013, there were 24 studies with both *p*-stats and *p*-value, 67 studies only with *p*-stats, six studies only with *p*-value, and 62 studies without *p*-stats or *p*-value. In 2018, there were 32 studies with both *p*-stats and *p*-value, 34 studies only with *p*-stats, 11 studies only with *p*-value, and 51 studies without *p*-stats or *p*-value. (B) Studies with either p-stats or *p*–value were used in the z-curve analyses. (C) Studies with *p*-stats were used in the *p*-curve analyses (Supplemental 4).

*Brunner & Schimmack, 2020*). As mentioned above, we mostly report the results of *z*-curve analyses (the analysis plan and results of the *p*-curve analyses are reported in Supplemental 4).

### Z-curve analysis

With *z*-curve analysis, we estimated the mean "power" of a set of studies. In the context of *z*-curve analysis, "power" refers to "unconditional power," which is a long-run frequency of statistically significant results without conditioning on a true effect (*Bartoš & Schimmack, 2022*). As such, a study will have an unconditional power equal to the alpha (*e. g.*, 5%) when the null hypothesis is true. Since the power is a function of the effect size and the sample size, a small power indicates a small effect size, a small sample size, or both—a lack of evidential value at any rate.

Z-curve analysis capitalizes on the fact that *z*-values, derived from *p*-values, conform to a normal distribution when the *p*-values originate from a set of studies with the same power. The mean of this normal distribution corresponds to the power of those studies. Conversely, by identifying the normal distribution that fits the observed *z*-value distribution, we can estimate the power of the studies under examination. To be more precise, *z*-curve analysis first computes *z*-values by transforming reported *p*-values into two-tailed *z*-values, with a focus on the positive values. Then, it utilizes a finite mixture model, fitting a mixture of truncated folded normal distributions with differing means to the distribution of the *z*-values that exceed 1.96 (*i.e.*, *p*s < 0.05). These procedures address several challenges, including 1) the impossibility of ascertaining the direction of the effects tested within the target articles, 2) the likely distortion in the observed *z*-value distribution due to publication bias, and 3) the heterogeneity of power among the studies in question. Following the fitting of a truncated *z*-curve, it extrapolates the full distribution to cover the $0 < z < 1.96$ area. We recommend consulting Figure 1 in the *z*-curve 2.0 paper (*Bartoš & Schimmack, 2022*) to gain a better grasp of the core concepts with a visual representation.

Two types of "mean powers" can be estimated from the fitted *z*-curve. One is the expected discovery rate (EDR), which is the mean power of all studies with statistically significant and non-significant results. Put differently, it is an estimation of the percentage of significant studies among all studies that have ever been conducted, including both reported and not-reported studies. The second is the expected replication rate (ERR), which is the mean power of statistically significant studies. In other words, the ERR is the expected success rate of the exact replication ("time-loop" replication) of the target studies that produced $p < 0.05$ results. As exact replications are practically impossible in psychology, the ERR is an optimistic estimation of the success rate. Another important metric in the *z*-curve analysis is the observed discovery rate (ODR), which is the percentage of significant ($p < 0.05$) studies among all eligible studies included in the analysis.

We preregistered to follow *Schimmack (2020)* in our interpretation of *z*-curve analyses. However, several modifications were introduced during the review process. We describe our final approaches below. Any deviations from the preregistration are noted in the footnotes.

![PeerJ]

[3] We preregistered to see if the 95% CI of the ERR included the ODR. As a reviewer pointed out that such a direct comparison between the ERR and the ODR is inappropriate, we modified our analytic approach accordingly.

[4] This interpretation of ERR was not pre-registered but was introduced following a discussion with the reviewers. It substitutes the pre-planned test of evidential value by p-curve analysis.

[5] We did not pre-register to examine the file drawer ratio. Thus, our interpretation is inevitably *post hoc*.

[6] We pre-registered to use the second reported p-values/stats for the sensitivity analyses. However, it turned out that picking up the last p-values/stats required a much simpler R script, making the possibility of errors much smaller. Therefore, we decided to use the last p-values/stats for sensitivity analysis.

First, we examine the ERR. A common misconception is that a high ODR implies the robustness of positive findings. In fact, the ERR serves as a much better indicator of robustness. For instance, if we find an ERR of 0.5, it implies that only half of the positive results are likely to be successfully replicated, even in the presence of a considerably larger ODR.[3] Importantly, if the 95% confidence interval (CI) of the ERR does not include 5%, it indicates that the set of studies holds statistically significant, albeit small, evidential value, as the ERR should be 5% by chance alone.[4] The lower bound of the 95% CI represents the lower estimate of the evidential value.

Second, we compared the EDR and ODR. The EDR is an estimation of the proportion of significant studies among all studies that have been conducted, including both reported and not-reported studies. If there is no selective reporting by researchers, the EDR should be as high as the ODR. Consequently, if the EDR is significantly lower than the ODR, we would conclude that we have evidence that the authors selectively reported those studies with significant p-values while putting non-significant results away in their file drawers.

We can use EDR to estimate the maximum false discovery rate (FDR) with *Sorić*'s *(1989)* formula. FDR is the share of false-positive studies among the studies that have been reported to be statistically significant. Even though a higher FDR (*e.g.*, FDR > 0.5) does not necessarily indicate QRPs by the researchers, if it was ever observed, this suggests that something may be going wrong in the field. Therefore, we decided to report the FDR. In addition, we reported the file drawer ratio, which is computed by (1-EDR)/EDR. This indicates the relative size of the expected frequency of non-significant studies (1-EDR) to that of significant studies (EDR). Since not all non-significant results are left in the file drawer, nor all significant results are reported, the file drawer ratio can be either optimistic or conservative.[5]

We used p-values computed from p-stats for the z-curve analyses as far as p-stats were available. When a study did not report p-stats but reported a respective p-value, we included it in the analyses (Fig. 1). This enabled us to have relatively large power for the analysis. There are three methods for fitting a mixture of normal distributions to the observed z-value distribution (Kernel Density 1, Kernel Density 2, and expectancy-maximization; KD1, KD2, and EM, respectively). We are reporting results from the EM method as primary analyses and we report other results in Supplemental 5. Estimations were obtained using the latest z-curve package (*Bartoš & Schimmack, 2020*) for R (*R Core Team, 2022*), which was available at the time of analysis (z-curve 2.1.2).

### Sensitivity analysis

Several conference papers have reported more than two p-stats/p-values. We used the p-stats/p-values that appeared last in the papers for sensitivity analysis.[6] Thus, when there was only one eligible p-stats/value in a paper, we used it for the sensitivity analysis. If there were more, we would consider the last one.

Another set of sensitivity analyses would include z-curve analyses only with studies with p-stats, as we did with the p-curve analyses (Fig. 1). This was to determine if the evidential value reported by the z-curve analyses was as high as that reported by the p-curve analyses if we used the same eligibility criteria.

**Table 2 Z-curve analyses with papers in 2013 and 2018 (Primary analyses).** Results of *z*-curve analyses with *p*-stats or *p*-values in the 2013 and 2018 conference paper employing the EM algorithm for fitting. The Expected Replication Rate (ERR), Expected Discovery Rate (EDR), and Observed Discovery Rate (ODR) are presented with 95% confidence intervals. The False Discovery Rate (FDR) was computed using the formula proposed by *Sorić (1989)*, which estimates the maximum false discovery rate. The file drawer ratio is (1-EDR)/EDR, an estimation of relative file drawer size compared to the number of positive results.

| | 2013 | | | 2018 | | |
|---|---|---|---|---|---|---|
| | **Estimate** | **Lower** | **Upper** | **Estimate** | **Lower** | **Upper** |
| ERR | 0.31 | 0.16 | 0.48 | 0.44 | 0.22 | 0.62 |
| EDR | 0.07 | 0.05 | 0.15 | 0.13 | 0.05 | 0.57 |
| ODR | 0.76 | 0.66 | 0.84 | 0.64 | 0.52 | 0.74 |
| FDR (maximum) | 0.68 | 0.30 | 1.00 | 0.35 | 0.04 | 1.00 |
| File drawer ratio | 12.99 | 5.75 | 19.00 | 6.57 | 0.76 | 19.00 |

## RESULTS

### Z-curve analyses

We report the results of the *z*-curve analyses using the EM method and the first reported *p*-stats or *p*-values as primary analyses. To evaluate the evidential value of the set of target studies, we computed two types of mean powers: ERR and EDR.

### Studies in 2013

There were 97 *p*-stats/*p*-values from conference papers published in 2013. Among these, 74 were significant ($p < 0.05$); the ODR was 76% (Table 2). We converted the *p*-values into *z*-values and used 65 of them (which were smaller than six) for the *z*-curve fitting. We used all 74 *z*-values to estimate the mean powers. We found that the ERR was 31%, indicating that approximately one-third of exact replications of the positive findings are likely to be successful. Notably, the lower boundary of the ERR was larger than 5%, indicating that the set of studies had evidential value, albeit small.

The EDR—an estimation of the share of significant studies among all studies that have been conducted—was 7%, which was much smaller than the ODR (Table 2, Fig. 2). The 95% CI did not overlap with that of the ODR (5% to 15% and 66% to 84%, respectively). Note that the EDR should be as high as the ODR if there is no selective reporting by the authors. The file drawer ratio was 12.99, suggesting that there could be about 13 negative and not-reported results per one positive and reported result. These results revealed the selective reporting by the authors.

The maximum FDR was 68%, meaning that 68% of the positive findings could be false positives. The 95% CI was wide (0.30, 1.00) and *Sorić (1989)*'s formula returned the maximum estimates of FDR, limiting conclusive arguments.

A sensitivity analysis with the last *p*-stats/*p*-values reported in a paper showed a similar pattern (Table 3); both the ERR and the EDR were smaller than the ODR, and neither of their 95% CIs overlapped with that of the ODR. The maximum FDR suggests that, at most, approximately half of the positive findings were false positives. The size of the file drawer
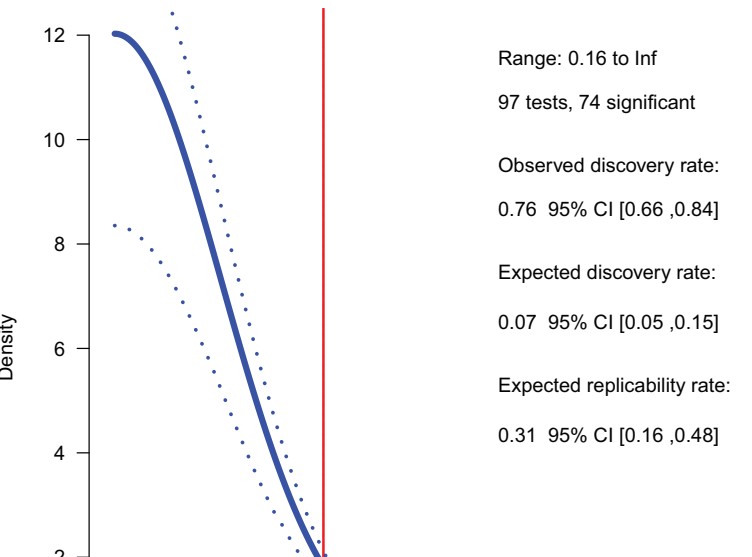

**Figure 2 Z-curve analysis of 2013 studies (EM method, first reported *p*-stats or *p*-values).** A histogram of *z*-values converted from the observed *p*-values is presented. *Z*-curve fitted to the histogram using the EM method (solid line) and its 95% confidence intervals (dashed lines) are depicted. *Z*-curves indicate the estimated distribution of *z*-values that would be observed if there was no publication bias (*i.e.*, selective reporting by the authors for the current study). The zcurve 2.1.2 package for R produced the graph (*Bartoš & Schimmack, 2020*).               

**Table 3 Z-curve analyses with papers in 2013 and 2018 (sensitivity analyses).**

|  | 2013 | | | 2018 | | |
|---|---|---|---|---|---|---|
|  | **Estimate** | **Lower** | **Upper** | **Estimate** | **Lower** | **Upper** |
| ERR | 0.35 | 0.19 | 0.54 | 0.21 | 0.05 | 0.39 |
| EDR | 0.10 | 0.05 | 0.32 | 0.07 | 0.05 | 0.24 |
| ODR | 0.66 | 0.56 | 0.75 | 0.56 | 0.44 | 0.67 |
| FDR (maximum) | 0.49 | 0.11 | 1.00 | 0.73 | 0.17 | 1.00 |
| File darwer ratio | 9.34 | 2.13 | 19.00 | 13.85 | 3.22 | 19.00 |

**Note:** The results of the *z*-curve analysis with *p*-stats/*p*-values reported last in each conference paper employing the EM method for fitting.

was estimated as nine times as large as that of the reported studies. Other sensitivity analyses using studies with either *p*-value or *p*-stats and the analyses with kernel density methods (KD1 or KD2) for fitting showed similar patterns (see Supplemental 5 for details).

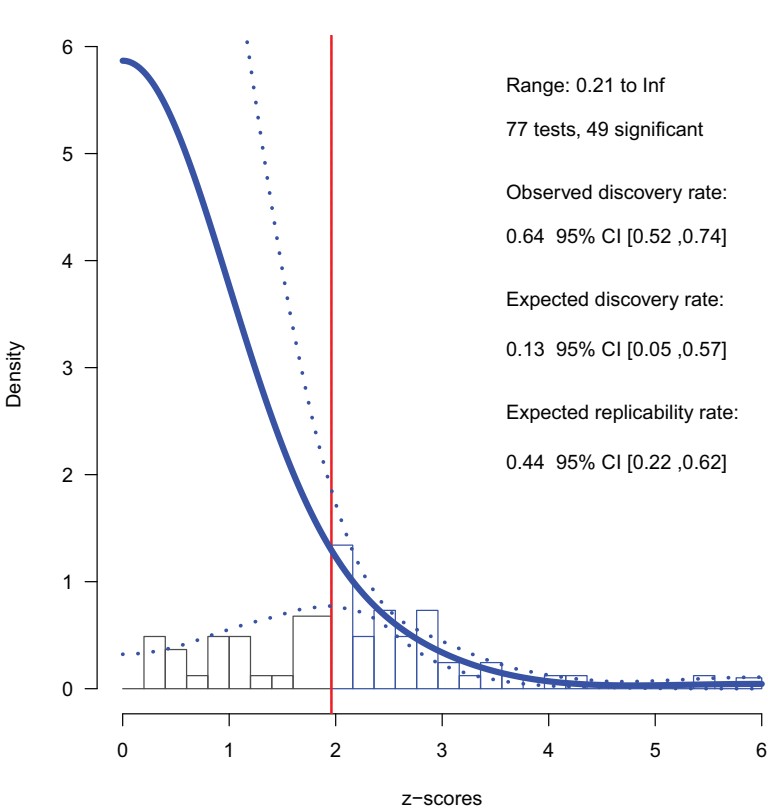

**z-curve (EM via EM)**

Range: 0.21 to Inf

77 tests, 49 significant

Observed discovery rate:

0.64  95% CI [0.52 ,0.74]

Expected discovery rate:

0.13  95% CI [0.05 ,0.57]

Expected replicability rate:

0.44  95% CI [0.22 ,0.62]

**Figure 3 Z-curve analysis of 2018 studies (EM method, first reported *p*-stats or *p*-values).** A histogram of *z*-values converted from the observed *p*-values is presented. The *z*-curve obtained using the EM method (solid line) is indicated with its 95% confidence intervals (dashed lines). The zcurve 2.1.2 package for R produced the graph (*Bartoš & Schimmack, 2020*). 

### Studies in 2018

In 2018, there were 77 *p*-stats/*p*-values from conference papers. Among them, 49 were significant ($p < 0.05$); the ODR was 64% (Table 3). We used 41 converted *z*-values smaller than six for the fitting and all 49 *z*-values for the final estimation (Fig. 3). The ERR was 44%, indicating that more than half of exact replications of the positive findings are likely to fail. The lower 95% CI of the ERR was larger than 5%, indicating a small but significant evidential value. The EDR was 13%, which was less than one-third of the ODR. The 95% CI [5–57%] included that of the ODR (52% to 74%). Since the CI of the EDR did not overlap with that of the ODR in 2013, improvements were observed. The file drawer ratio was 6.57, and the maximum FDR was 35%. That is, there could be about seven unreported negative results per one positive report, and about one-third of the reported positive findings could be false positives. Nevertheless, these figures were about half of what they were in 2013 (file drawer ratio = 12.99, FDR = 68%), indicating improvements. Overall, the 2018 studies had certain evidential value and showed a slight improvement from 2013. Even so, evidence of selective reporting by the authors was still suggested, and the power was not strong in an absolute sense.

A sensitivity analysis with the last $p$-stats/$p$-values reported in a paper revealed a similar but slightly different pattern (Table 3). The ERR and EDR were smaller (21% and 7%), and the 95% CI of the EDR did not overlap with that of the ODR (5% to 24%, and 44% to 67%, respectively). The maximum FDR was large (73%), implying that about three-quarters of the positive findings may have been false positives. (See Supplemental 5 for details on the sensitivity analysis).

Overall, the ERR in the sensitivity analysis was lower than that in the primary analysis. The 95% CI of the EDR overlapped with the CI of the ODR in the primary analysis but not in the sensitivity analysis. In terms of the point estimates, the EDRs were considerably smaller than the ODRs in both the primary and sensitivity analysis.

## DISCUSSION

We conducted a systematic review of conference papers in social psychology from two large Japanese psychological societies, the JPA and the JSSP, in 2013 and 2018. We analyzed the distributions of $p$-values from experimental studies with a directed prediction using $z$-curve (*Bartoš & Schimmack, 2022*; *Brunner & Schimmack, 2020*) and $p$-curve analyses (*Simonsohn, Nelson & Simmons, 2014a*, *2014b*; *Simonsohn, Simmons & Nelson, 2015*; Supplemental 4). Specifically, we examined whether Japanese social psychologists engaged in selective reporting, even when pressure from reviewers and editors was absent (*i.e.*, conference papers). In addition, we explored whether there was a change in the evidential value of the reported findings under pressure-free conditions between 2013 and 2018 when the replicability crisis was actively advertised in the Japanese psychological community.

### Selective reporting and changes between 2013 and 2018

The $z$-curve analyses suggest selective reporting by the authors of the conference papers. The EDR (which should be as high as the ODR if all studies were reported) was much lower than the ODR in 2013 (7% *vs*. 76%, respectively). The 95% CI of the EDR (5% to 15%) did not include that of the ODR. Selective reporting would explain the discrepancy between the EDR and the ODR. The file drawer ratio, (1-EDR)/EDR, was 12.99, suggesting that there could have been 13 unreported negative results per one reported positive finding. It should be noted that the estimate might contain considerable error because not all negative results were left in the file drawer; nor were all positive results necessarily reported.

We observed a similar pattern in 2018. The point estimate of the EDR was much smaller than the ODR (13% *vs*. 64%), and the file drawer ratio was as high as 6.57. Substantial numbers of negative results could have been left in the file drawer. Notably, however, the 95% CI of the EDR overlapped with that of the ODR, and the file drawer ratio was nearly halved from 2013. Thus, we would conclude that the evidence for selective reporting was apparent in 2018, but its magnitude was smaller than in 2013.

It may be interesting to compare the magnitude of selective reporting under the peer-pressure-free condition (current study) and that in journal-published papers.[7] *Schimmack (2020)* estimated the ERR and the EDR among published social psychology articles by

---

[7] We did not pre-register the comparison of selective reporting between the conference papers and the journal papers. We introduced it during the review process.

analyzing data set by *Motyl et al. (2017)*, who had a representative sample of articles published in the *JPSP*, *PSPB*, *JESP*, and *PS* in 2003–2004 and 2013–2014. Specifically, *Schimmack (2020)* analyzed the 678 articles reported in the *JPSP*, *PSPB*, or *JESP* with suitable statistics for *z*-curve analysis. The estimation of the EDR was 19% (95% CI [6%, 36%]). The point estimate was larger than that for the current study (7% in 2013 and 13% in 2018), although the CIs overlapped in both years. The file drawer ratio was 4.26, about one-third of that in 2013 (12.99) and two-thirds of that in 2018 (6.57) for the current study (Table 2). Evidence of publication bias seemed stronger among the peer-pressure-free conference papers. However, it is also probable that because the conference paper authors did not have to care about the evaluation by peer researchers, they were more likely to present challenging and preliminary ideas and results at the conference venues. If this were the case, it would not be surprising to find lower EDR among the conference papers, which led to a higher file drawer ratio: (1-EDR)/EDR.

It is more important to look at the discrepancy between the EDR and ODR: the frequency of positive results among all reported results. For the journal papers, this figure was 71% (90% minus 19%). For the conference papers, they were 69% (76% minus 7%) in 2013 and 51% (64% minus 13%) in 2018. Notably, the biases were comparable between the journal papers (which were published in 2003–2004 and 2013–2014) and the conference papers in 2013. This may indicate that, as far as statistical significance is concerned, there used to be no additional publication bias imposed by journal editors and reviewers in those years. However, it may be inappropriate and nonsensical to naively compare journals under mostly Western cultural influence with conferences under Japanese cultural influences, no matter how similar they may appear to be.

We cannot specify the criteria through which the authors selected the results (*e.g.*, novelty, surprisingness, or statistical significance) and how they engaged in it (*e.g.*, *p*-hacking, HARKing). We can only say that there were more positive outcomes than expected from the distribution of *p*-values. Likewise, we cannot discern the psychology of the psychologists who resorted the selective reporting. They may have internalized the cultural norm of the field to value statistical significance and simply disregard non-significant results as meaningless. They might not want to report negative outcomes that would contradict their past or future published works, which would mainly contain positive results. If the latter were the case, selective reporting could have occurred without internalizing the field's cultural norms.[8]

[8] Our reviewers (Dr. Schimmack and Dr. Syrjänen) suggested this possibility.

## Evidential value and changes between 2013 and 2018

The *z*-curve analyses showed that the set of studies in both years had evidential values. The ERR, which estimates the mean power of positive studies, was significantly larger than 5% in 2013 and 2018. As ERR can be 5% by chance alone, it indicated that positive results were not entirely false positives.

We noted improvements in the mean power between 2013 and 2018. In 2013, the ERR was only 31% and the 95% CI [16–46%]. Even if we could conduct exact (time-loop) replications of positive findings, the success rate was expected to be less than 50% at most. The figure was much lower than one might erroneously anticipate when considering the

ODR (76%) as an indicator of evidential value. The evidential value of the conference papers in 2013 was weak. However, the figures displayed some improvements in 2018. The point estimate of the ERR increased (44%), and the upper bound of the 95% CI [22–62%] was close to the ODR (64%). However, the 95% CIs in 2013 and 2018 overlapped (16% to 46% *vs.* 22% to 62%), failing to indicate the difference between the two sample years. This may be due to the small sample sizes.

The maximum FDR improved from 2013 to 2018; it was 68% (95% CI [30–100%]) in 2013, implying that more than half of the positive findings may have been false positives. It became 35% (95% CI [5–100%]) in 2018, almost halved from 2013, hinting at an improvement in evidential value. Still, there remains the possibility that about one-third of the positive findings were false positives.

Results of the *p*-curve analyses were consistent with the *z*-curve analyses (Supplemental 4). They showed evidential values of the target studies in 2013 and 2018 and suggested improvements during the 5 years.

Overall, we can conclude that there was a slight improvement in the evidential value between 2013 and 2018. However, the weak mean power (ERR) and the large maximum FDR still highlighted weak evidential value and did not warrant optimism.

## CONCLUSIONS

There is a clear indication of selective reporting in social psychology conference papers presented at two Japanese psychology conferences in 2013 and 2018. That is, Japanese social psychologists kept non-significant *p* > 0.05 results in a file drawer and selectively reported *p* < 0.05 outcomes. Notably, this happened under a condition where the researchers had almost complete freedom to report anything without caring for the reviewers' eyes.

This kind of QRP inevitably weakens the evidential value of the reported findings, as shown by the low expected successful replication rate, which was less than 50%. However, this does not necessarily mean that researchers resorted to more intense forms of QRPs, such as *p*-hacking, and covered the conference venue with the "null field" (*Ioannidis, 2005*). We found that the set of studies had evidential value, albeit small. At the same time, though, a substantial proportion of the positive results could have been false positives, which hinders optimism.

We also observed slight changes in the behavior of the researchers between 2013 and 2018. In general, the 2018 studies demonstrated stronger evidential value than the 2013 studies. The ERR increased and the maximum FDR decreased. Evidence indicating selective reporting is also weaker. Again, these changes occurred under conditions without enforcement by the editors and reviewers. Japanese social psychologists may have internalized the new community's norms to avoid engaging in selective reporting (*Vazire, 2019*).

While the improvement in recent years is welcome, selective reporting was present even in 2018 and the evidential value of the studies was small in an absolute sense. When we refer to and cite papers published in reviewed journals over the years, we need to consider the possibility that the evidential value of a study may have been damaged by selective

reporting by the authors and probably by additional publication bias in the peer review process (*Franco, Malhotra & Simonovits, 2014*).

Present study has some limitations. First, we cannot determine if novelty and surprisingness affected the researchers' decisions on which study to report at the conference. Although we suspected that psychologists tended to selectively report novel, surprising, and statistically significant results, we could only test part of the hypothesis, *i.e.*, if researchers selectively reported statistically significant results. Second, the findings cannot and should not be naively generalized. We looked only at social psychology studies presented at Japanese academic conferences in 2013 and 2018. The targets were restricted to studies with experimental manipulations and directed hypotheses. We believe that we have presented evidence that social psychologists in Japan used to voluntarily engage in selective reporting without direct peer review pressure, at least until approximately 5 years ago. Still, we do not claim that the same should always be true in other times, countries, or disciplines, in the same way as we cannot generalize findings from WEIRD (Western/ White, Educated/English, Industrialized, Rich, and Democratic) samples to humanity in general (*Cheon, Melani & Hong, 2020*; *Henrich, Heine & Norenzayan, 2010*).

## ACKNOWLEDGEMENTS

The systematic review protocol has been registered. The anonymized Stage 1 protocol is depicted in Supplemental 1 and the signed preregistration version can be found at https:// osf.io/4vxkb. The anonymized Stage 2 protocol is displayed in Supplemental 2 and the signed preregistration version can be found at https://osf.io/mn5f4. We would like to thank Kazuaki Hatadani for collecting and archiving the PDF files of the conference papers, thus establishing the environment for target article coding. We would also like to express our sincere gratitude to the editor and the reviewers for their insightful guidance and valuable feedback throughout the review process.

### Funding

This study was supported by JSPS KAKENHI Grant Number 19H01750. The funders had no role in study design, data collection and analysis, decision to publish, or preparation of the manuscript.

### Grant Disclosures

The following grant information was disclosed by the authors:
JSPS KAKENHI: 19H01750.

### Competing Interests

Kai Hiraishi, Asako Miura, Masataka Higuchi, and Yoshitsugu Fujishima are members of the JPA and the JSSP. Masaki Suyama is a member of the JSSP.

## Author Contributions

- Kai Hiraishi conceived and designed the experiments, analyzed the data, prepared figures and/or tables, authored or reviewed drafts of the article, and approved the final draft.
- Asako Miura conceived and designed the experiments, analyzed the data, authored or reviewed drafts of the article, and approved the final draft.
- Masataka Higuchi conceived and designed the experiments, analyzed the data, authored or reviewed drafts of the article, and approved the final draft.
- Yoshitsugu Fujishima conceived and designed the experiments, analyzed the data, authored or reviewed drafts of the article, and approved the final draft.
- Daiki Nakamura conceived and designed the experiments, performed the experiments, analyzed the data, authored or reviewed drafts of the article, and approved the final draft.
- Masaki Suyama conceived and designed the experiments, performed the experiments, analyzed the data, authored or reviewed drafts of the article, and approved the final draft.

## Data Availability

The data and R-script are available at OSF: Hiraishi, Kai, Asako Miura, Masataka HIGUCHI, Yoshitsugu Fujishima, Daiki Nakamura, and Masaki Suyama. 2023. "Data and Scripts for Systematic Review of Conference Papers at JPA and JSSP." OSF. December 21. DOI 10.17605/OSF.IO/KDZTX.

## Supplemental Information

Supplemental information for this article can be found online at http://dx.doi.org/10.7717/peerj.16763#supplemental-information.

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
