# Peer review of "A systematic review of conference papers presented at two large Japanese psychology conferences in 2013 and 2018: did Japanese social psychologists selectively report p < 0.05 results without peer review?"

_PeerJ, doi:10.7717/peerj.16763_

## Round 0.1 · original submission · Major Revisions

After reviewing the comments from the reviewers, I have made the decision to request a major revision of your manuscript before resubmitting.

Although the reviewers find your contribution interesting and potentially publishable, they have all identified important weaknesses that currently prevent its publication.

Reviewer #3 highlights a significant concern regarding the lack of comparison with peer-reviewed studies, which weakens the conclusions. Additionally, Reviewers #1 and #3 offer an intriguing alternative interpretation of your results. They suggest considering the possibility that the choice to submit a conference poster may indicate a willingness to pursue a publication.

Both Reviewer #2 and Reviewer #3 recommend providing a more thorough introduction to the methods used. This would benefit the majority of readers who may not be familiar with the z-curve or the p-curve. Considering the criticisms raised by Reviewer #1 and Reviewer #2, I also question whether it would be reasonable to exclude the p-curve from the paper and focus solely on the z-curve.

Reviewers #1 and #2 have also identified relevant flaws in the methodology employed. I strongly advise addressing these concerns by implementing their suggested improvements.

Please notice that this was meant to be just a summary of the Reviewers' point, but I strongly recommend to address all their concerns point-by-point.

·

Basic reporting

see general comments

Experimental design

see general comments

Validity of the findings

see general comments

Additional comments

The idea to examine changes in research practices using statistical tools like p-curve and z-curve is a good one. The focus on conference presentations is interesting because conference presentations are not very competitive (posters). However, researchers typically will try to publish their work later on. Thus, selection bias is expected. It might be interesting to compare bias in conference presentations to bias in journal publications, but in general I think it is well known that selection bias starts in the lab.
The use of p-curve and z-curve may be interesting for readers who are familiar with p-curve, but not with z-curve. However, p-curve results of evidential value are redundant with the more precise z-curve estimate of the replication rate. If the 95% CI does not include 5% (expected by chance alone), the studies as a set have evidential value. If the 95%CI does not include 33% and is above that value, power is at least 33%, but we can just look up the estimated value to get more precise information. P-curve does not make it possible to compare results for 2013 and 2018, while z-curve provides confidence intervals that can be used for a direct comparison and significance test (non-overlapping 95%CI imply significance at about alpha = .01). I therefore focus on the z-curve results.

Unfortunately, the authors seem to have excluded z-scores greater than six form the analysis. This might be based on a misinterpretation of the method. The method does not use these values for estimation but they need to be included to compute the proper estimates for all values. Z-curve assumes that they have 100% power. So, if there were 74 significant results and 65 with z-scores below 6, 9 z-scores were omitted. So, if the ERR is estimated to be 31%, the correct ERR would be( .31*65 + 1*9)/74 = .39. Not a big difference but I would suggest to rerun the analysis with all non-significant and significant results. In any case, 31% or 39% is a very low estimate of replicability.
To examine selection bias, it is necessary to compare the ODR with the expected (!) replication rate. The EDR is an estimate of the full distribution, including non-significant results.
The CI estimate of the EDR is surprisingly tight given the relatively small sample size. This might be due to the omission of the highly significant (z > 6) results. However, it is likely that the new results will still show clear evidence of selection bias (i.e., upper limit of EDR is lower than lower limit of ODR).

The results for 2018 also need to be recomputed and may show that the ODR falls into the 95%CI of the EDR. Of course, this does not mean that there is no selection bias, but given the sample size the evidence is not strong enough to show it. I think more important is a comparison of the point estimates with some cautious interpretation. Stronger evidence would require coding more conference presentation to show time trends in ODR, EDR, and ERR.

I did not understand what first and last meant in Tables 2 and 3. A single table with results for 2013 and 2018 with estimates for the full dataset would be easier to interpret.

Overall, the results show no evidence that research practices have changed notably from 2013 to 2018. Maybe 2018 is too early, but the results are consistent with some of my own unpublished results that notable changes are limited to specific journals where editors enforced new norms.

I would be happy to help with z-curve analysis or interpretation of the results.

Best, Ulrich Schimmack



p-stats/p-values just say p-values

·

Basic reporting

I am guessing many readers will be unfamiliar with z-curve analysis, if not p-curve analysis, and so I think a more in-depth explanation of both techniques and the underlying logic of both of them than what is already present would be useful for potential readers to comprehend the entire study.

The authors' description of the process by which studies are selected for inclusion at conferences is also minimal, and I do not feel like it acquaints the reader well with the process. I am referring specifically to the section in the Introduction called "Peculiarities of the conference paper format of Japanese psychological societies." Because the distinction between the peer review process at journals and the (non-) peer review process for conferences is the whole basis for the rationale behind this study, it would behoove the authors to describe how exactly studies are accepted to conferences.

Another thing I believe the authors should report from the results of their z-curve analyses is the file drawer ratio. Given their interest in selective reporting, I would think that this would be an important estimate for them to report and discuss.

In the z-curve plots (i.e., Figure 5 and Figure 6), it is helpful to include the annotations in those figures. When performing a z-curve analysis, one can request annotations on the figures that will display the ODR, EDR, and ERR alongside the figure. It is not essential, but it is helpful if a reader wants to acquire that information quickly.

The authors claim that z-curve yields two types of average power--the estimated replication rate (ERR) and the estimated discovery rate (EDR). This is not accurate. The ERR is the estimated average power of all the original studies, while the EDR is the percent of the original studies one would expect to have been statistically significant in light of the estimated average power.

On the subject of the outcome of the z-curve, the authors discuss comparing the ERR with the observed discovery rate (ODR). They claim that comparing these two is a critical way of assessing the degree to which selective reporting is present in the studies included in the z-curve. While their rationale here makes sense, it is really the EDR one uses in this comparison. If the EDR is substantially smaller than the ODR--and in particular if the confidence interval for it does not include the ODR's confidence interval--then this suggests selective reporting. The authors are aware of this, as they also discuss comparing the EDR and ODR, but I think they should delete their sections about comparing the ERR and ODR, as such a comparison is unorthodox when reporting the results of a z-curve analysis.
The authors also claim that z-curve and p-curve are kinds of meta-analyses, which is not strictly speaking accurate. While they are meta-analytic techniques and are often used alongside a meta-analysis, they do not necessarily do what a traditional meta-analysis does in estimating an average effect size across the included studies. While one can use p-curve to run a meta-analysis if all of the p-values entered come from the same effect being tested, one cannot do this in z-curve. And, in any case, the authors do not meta-analyze the studies they include--with good reason, as the studies do not all test the same effect. Thus, it would be much more accurate if the authors did not use meta-analysis in reference to their current work. They might consider referring to it as a systematic review.

Experimental design

I think the authors should consider leaving out the p-curve analysis altogether. There is mounting evidence that z-curve analysis is much more accurate in estimating both replicability and selective reporting than p-curve analysis. For example, Brunner and Schimmack (2020) find that p-curve frequently overestimates population mean power relative to z-curve. As the authors own findings show, the results of p-curve analysis tend to be overly optimistic, and so might give potential readers the wrong idea about the state of research in the field.

Validity of the findings

The authors' tone about their findings seem overly optimistic when they compare the 2013 results to the 2018 results. They write in their conclusion as though there has been a vast improvement from 2013 to 2018. While their findings do show a degree of improvement, the findings for the 2018 studies were still harrowing. An ERR of 44%--or as low as 21% as per the authors' sensitivity analysis--and a false discovery risk of 35% and a discrepancy of 51 percentage points between the EDR and the observed discovery rate would not be a favorable outcome for a z-curve if one assessed them without the 2013 studies for comparison. It means that less than half of studies are predicted to replicate, that selective reporting is still substantial, and that up to a third of the studies could be false positives. I think the authors should revise their conclusion to be clearer about how big the issues with replicability and selective reporting of results still are in the field.

Additional comments

I believe this to be a well-founded and interesting study that adds to existing knowledge about replicability issues. The idea to run replicability analyses using z-curve and p-curve on conference papers makes for an interesting way of assessing the degree to which selective reporting of results really has to do with pressures from journals or from researcher practice. This study suggests that researcher practices have just as much to do with selective reporting as pressures from journal editors. Comparing studies from the year 2013 and the year 2018 also makes for an interesting look at the degree to which replicability and selective reporting have (or have not) improved. The authors’ results suggest that there has been slight improvement since 2013, but estimated replicability rates are still low and evidence of selective reporting is still high in the 2018 studies. This puts their findings in line with other recent studies investigating the changes in replicability and selective reporting since the replicability crisis became more well-known, which tend to show that rates of replicability and selective reporting have improved, but are still far from ideal.

·

Basic reporting

I like the idea/concept of investigating publication bias in non-peer-reviewed published studies, a naturalistic experiment. The lack of comparison with peer-reviewed studies weakens the results/conclusions. However, the idea is nonetheless good and could inspire future meta-scientific research.
The manuscript is well-written and easy to read for a broad audience. The background and context are sufficiently covered with a focus on psychology/social psychology. However, it is well known that publication bias is present in most fields that use statistical inference. The manuscript could benefit from a broader background on publication bias, followed by motivation on why publication bias in social psychology is particularly interesting.
The authors might mention somewhere around line 100 in the manuscript that p- or z-curve analyses measure how well the distribution of results matches the expected distribution of results without publication bias for readers unfamiliar with these methods.
I am delighted with the authors’ open research practices, preregistered analyses, and sharing of raw data and code (very well-commented and easy to understand).

Experimental design

The results and conclusions of this manuscript could have been significantly strengthened by comparing published studies in peer-reviewed conferences. However, this might not be viable as the analyses were pre-registered.

Validity of the findings

Although I’m sympathetic to the notion that there are norms that are more or less internalized (line 403 in manuscript), however, selection for publication bias might happen even without peer review just because many authors present preliminary results at conferences followed by a peer-reviewed journal article. A published conference article can inflate the citation count (self-citation), and many authors might be unwilling to present conflicting results from the same dataset. Thus, there are incentives for QRPs even in the absence of peer review.

---

## Round 0.2 · Major Revisions

With the reviewers' comments in hand, I recommend you revise and resubmit the manuscript. Even though Reviewer #3 is satisfied with your changes, Reviewer #2 pointed out some remaining important inaccuracies that need to be fixed (most importantly, comparing EDR and ODR), and provided some useful suggestions on the relevant literature to rely on for clarifying your points.

·

Basic reporting

See Additional Comments.

Experimental design

See Additional Comments.

Validity of the findings

See Additional Comments.

Additional comments

I appreciate the improvements the authors made to this paper. In particular, they removed the p-curve from the paper proper, expanded their discussion of the peculiarities of how studies are selected for conferences in Japan, included the file drawer ratio in their results, included annotations in their z-curve plots, removed references to z-curve and p-curve as types of meta-analyses, and revised the tone of their conclusions pertaining to the change in the quality of research between 2013 and 2018. As far as my original comments went, these changes have improved the paper significantly.

However, there is one area in which I still find the work inadequate, and that is in how it describes z-curve analysis and the interpretation of the results of a z-curve analysis. The authors did not correct the comparison between the ERR and ODR they made in the original work. As I pointed out in my first review, the critical comparison is between the EDR and the ODR, and the ERR is interpreted on its own. Further, their description of what z-curve has many statements that are unclear or even inaccurate. I might recommend the authors read Sotola and Crede (2022), in particular the section on z-curve analysis in the Introduction and Method sections—ps. 897-898 and p. 901. I think this paper provides a good model for how to describe z-curve analysis accurately and clearly for readers who might be unfamiliar. I have elected to

Sotola, L. K., & Credé, M. (2022). On the predicted replicability of two decades of experimental research on system justification: A z-curve analysis. European Journal of Social Psychology, 52, 895-909. https://doi.org/10.1002/ejsp.2858

·

Basic reporting

I'm satisfied with the changes the authors made to the manuscript.

Experimental design

The argumentation is strengthened by the comparison with previously reported z-curve analyses of social psychology journals, i.e., Schimmack (2020).

Validity of the findings

The authors have expressed my concerns on how to interpret the results.

Additional comments

Good job; I'm happy with the manuscript in its current form.

---

## Round 0.3 · accepted · Accept

Dear Prof. Hiraishi,
Because of the comments of the Reviewer, I am glad to let you know that your publication is now suitable for publication on PeerJ.

Best,
MTL

·

Basic reporting

No comment

Experimental design

No comment

Validity of the findings

No comment

Additional comments

I think the authors have adequately addressed my concerns, and I believe this paper adds an important component to the literature on replicability.